# The ω-3 Polyunsaturated Fatty Acids and Oxidative Stress in Long-Term Parenteral Nutrition Dependent Adult Patients: Functional Lipidomics Approach

**DOI:** 10.3390/nu12082351

**Published:** 2020-08-06

**Authors:** Vit Kosek, Marie Heczkova, Frantisek Novak, Eva Meisnerova, Olga Novákova, Jaroslav Zelenka, Kamila Bechynska, Nikola Vrzacova, Jiri Suttnar, Alzbeta Hlavackova, Helena Dankova, Miriam Bratova, Nikola Daskova, Hana Malinska, Olena Oliyarnyk, Petr Wohl, Hana Bastova, Jana Hajslova, Monika Cahova

**Affiliations:** 1University of Chemistry and Technology, 166 28 Prague, Czech Republic; Vit.Kosek@vscht.cz (V.K.); jaroslav.zelenka@vscht.cz (J.Z.); Kamila.Bechynska@vscht.cz (K.B.); Nikola.Vrzacova@vscht.cz (N.V.); jana.hajslova@vscht.cz (J.H.); 2Center of Experimental Medicine, Institute for Clinical and Experimental Medicine, 140 21 Prague, Czech Republic; mahz@ikem.cz (M.H.); hedn@ikem.cz (H.D.); mihb@ikem.cz (M.B.); nikola.daskova@ikem.cz (N.D.); hana.malinska@ikem.cz (H.M.); ooliyarnyk@yahoo.com (O.O.); 34th Department of Internal Medicine, 1st Faculty of Medicine and General University Hospital, Charles University, 128 08 Prague, Czech Republic; fnova@lf1.cuni.cz (F.N.); eva.meisnerova@lf1.cuni.cz (E.M.); 4Department of Physiology, Faculty of Science, Charles University in Prague, 128 00 Prague, Czech Republic; olga.novakova@natur.cuni.cz; 5Academy of Sciences of the Czech Republic, Institute of Physiology, 142 20 Prague, Czech Republic; 6Institute of Hematology and Blood Transfusion, 128 20 Prague, Czech Republic; jiri.suttnar@uhkt.cz (J.S.); alzbeta.hlavackova@uhkt.cz (A.H.); 7Diabetology Center, Institute for Clinical and Experimental Medicine, 140 21 Prague, Czech Republic; pewo@ikem.cz (P.W.); habs@ikem.cz (H.B.)

**Keywords:** home parenteral nutrition, intestinal failure, fish oil, olive oil, oxidative stress, lipidomics, hydroxy-fatty acids, plasmalogens, lipidomics

## Abstract

Omega-3 polyunsaturated fatty acids (ω-3PUFAs) are introduced into parenteral nutrition (PN) as hepatoprotective but may be susceptible to the lipid peroxidation while olive oil (OO) is declared more peroxidation resistant. We aimed to estimate how the lipid composition of PN mixture affects plasma and erythrocyte lipidome and the propensity of oxidative stress. A cross-sectional comparative study was performed in a cohort of adult patients who were long-term parenterally administered ω-3 PUFAs without (FO/–, *n* = 9) or with (FO/OO, *n* = 13) olive oil and healthy age- and sex-matched controls, (*n* = 30). Lipoperoxidation assessed as plasma and erythrocyte malondialdehyde content was increased in both FO/– and FO/OO groups but protein oxidative stress (protein carbonyls in plasma) and low redox status (GSH/GSSG in erythrocytes) was detected only in the FO/– subcohort. The lipidome of all subjects receiving ω-3 PUFAs was enriched with lipid species containing ω-3 PUFAs (FO/–˃FO/OO). Common characteristic of all PN-dependent patients was high content of fatty acyl-esters of hydroxy-fatty acids (FAHFAs) in plasma while acylcarnitines and ceramides were enriched in erythrocytes. Plasma and erythrocyte concentrations of plasmanyls and plasmalogens (endogenous antioxidants) were decreased in both patient groups with a significantly more pronounced effect in FO/–. We confirmed the protective effect of OO in PN mixtures containing ω-3 PUFAs.

## 1. Introduction

Parenteral nutrition (PN) provides life-saving nutritional support in situations where caloric supply via the enteral route cannot cover the necessary needs of an organism [1]. The first well-tolerated lipid emulsion (Intralipid) was based on soybean oil rich in Omega-6 polyunsaturated fatty acids (ω-6 PUFAs) [2]. Besides their undeniable benefits, ω-6 PUFA administration was associated with inflammation and oxidative stress [3]. Since then, lipid emulsions containing pure olive oil (OO), pure fish oil (FO) or various blends of soybean oil (SO), OO, medium-chain triacylglycerols (MCT) and FO have been available. The introduction of FO into nutrition mixtures is associated with several beneficial effects due to their anti-inflammatory and immunomodulatory potential as eicosapentaenoic acid- (EPA, 20:5 *n*-3) or docosahexaenoic acid (DHA, 22:6 *n*-3)-derived eicosanoids, and docosanoids exhibit anti-inflammatory effects [4,5,6]. On the other hand, due to the high number of double bonds, ω-3 PUFAs may be more susceptible to lipid peroxidation and may increase the risk of oxidative stress [7]. Lipid mixtures with high contents of monounsaturated fatty acids (like OO) are at a lower risk of lipid peroxidation [8].

Studies examining the effects of different fatty acid constituents of lipid mixtures on oxidative stress differ both in the experimental setting and in the measured outcomes. Most of the in vivo studies addressing the issue of beneficial or adverse effects of FO-based nutrition mixtures were performed on a short-term (up to 14 days) basis. As far as oxidative stress is concerned, no effect of FO-based emulsions was reported in six studies [9,10,11,12,13,14] while their anti-oxidative effect was found in two studies [15,16]. One human study reported the pro-oxidative effect of FO emulsions [17]. In animal studies, two authors confirmed the anti-oxidative effect of FO in guinea pigs [18] or rats subjected to intestinal ischemia [19]. In contrast, Lavoie et al. described the increased oxidative stress in the lung tissue of guinea pigs administered FO (SMOFlipid) compared with SO (Intralipid) [20].

To our knowledge, only three studies addressed the long-term effect of FO-containing lipid emulsions on oxidative stress in humans. Zhang et al. reported that the switch from SO/MCT to FO-based lipid emulsion in a Chinese children cohort resulted in the improvement of liver function tests, inflammation and oxidative stress-related markers [21]. Pironi et al. described a cohort of adult patients with intestinal failure on long-term (˃3 months) home PN based either on OO/SO or SO/MCT/OO/FO lipid emulsions and found no difference in the FA trans-isomer content (free radical stress marker) [22]. Similarly, Klek at al. showed no difference in antioxidant enzyme activities when comparing SO with SO/MCT/OO/FO based lipid emulsions [23].

Several previous studies showed that the lipidome composition determines the susceptibility of lipid structures to potential deteriorating insults [24]. Thus, determining qualitative and quantitative changes in the lipidome may serve as a biomarker of inflammation or oxidative stress-related injury and/or identify possible risks. Given that we had found no such in-detail lipidomic study in patients on long-term PN in literature, we performed the observational study in the chronic intestinal failure population. The purpose of this research is to describe the effects of parenteral FO administration alone or in combination with OO and to assess its impact on redox state. This study aims (i) to assess the risk of oxidative stress associated with ω-3 PUFAs; (ii) to evaluate the capacity of OO to alleviate oxidative stress; and (iii) to explore plasma and erythrocyte lipidome of PN-dependent patients to find potential oxidative stress biomarkers with respect to the prevailing source of fat in PN emulsion.

## 2. Material and Methods

### 2.1. Patients and Controls

The study was carried out in a cohort of adult patients with chronic intestinal failure dependent on PN [25]. The inclusion criteria were: duration of PN ≥6 months; PN infusion ≥4 times per week; PN schedule and drug therapy unchanged during 4 months prior to the inclusion into the study. Exclusion criteria were active cancer, intestinal failure associated liver disease and refusal to participate. Blood samples were obtained 4–6 h after discontinuation of the overnight PN infusion. The detailed procedure of blood sample processing is described in Appendix A. Twenty-two eligible patients who were administered FO were recruited. Nine patients did not receive OO in lipid mixture (FO/–) while 13 patients were provided OO (FO/OO), which represented approx. 25% of total lipid intake, see Appendix A. The amount of glucose, amino acids and lipids per day were calculated from the total amount administered per week. Healthy controls (*n* = 30) were recruited from BMI- and age-matched individuals. The patient and control groups consisted of age-, gender- and BMI-matched subjects. These cohorts did not significantly differ in any tested parameter apart from serum LDL cholesterol, which was lower compared with controls, and serum CRP that was elevated in both FO/– and FO/OO groups (Table 1). Except for the composition of PN mixture, the patient groups were similar in age, sex, BMI, liver function test and lipid metabolism characteristics.

The study was conducted following Helsinki declaration and GCP. All patients and controls signed an informed consent approved by the respective Ethical Committees at each center (G 14-08-53).

### 2.2. Parameters of Oxidative Stress

Levels of reduced (GSH) and oxidized glutathione (GSSG) were assayed with the Glutathione in Whole Blood—HPLC detection kit (Chromsystems, Gräfelfing, Germany) according to the manufacturer instructions. Total antioxidant capacities (TAC) of plasma and erythrocyte lysates were determined by the oxygen radical absorbance capacity (ORAC) method using dipyridamole as described in Iuliano et al. [26]. The activity of Cu/Zn-superoxide dismutase (Cu/Zn-SOD) and glutathione peroxidase (GSH-Px) was determined using kits from Cayman Chemicals, MI, USA.

### 2.3. Determination of Malondialdehyde Concentration in Plasma and Erythrocytes

The total malondialdehyde (MDA) concentration in erythrocyte lysates and plasma was determined using stable isotope dilution assay based on liquid chromatography-tandem mass spectrometry, for details see Appendix A.

### 2.4. Lipidomics Analysis and Lipid Identification

The lipidome analysis was performed using a UHPLC system coupled to a quadrupole time-of-flight (QTOF) mass spectrometer. The collected data were processed with LipidMatch software (DOI 10.1186/s12859-017-1744-3) which uses MZmine 2 for feature extraction and an R script for lipid identification based on in-silico fragmentation databases. The whole procedure is described in detail in Appendix A.

### 2.5. Statistical Analyses

Statistical analysis of clinical data and parameters related to oxidative stress was performed using univariate statistics, namely Kruskal–Wallis test followed by Dunn’s multiple comparison test, correlation analysis (Spearman rank-correlation coefficient) and fold change analysis performed in GraphPad Prism. The power analysis was performed using relevant variability data for plasma MDA and SOD and assuming a two-sided 5% significance level. The sample size of four was considered sufficient to ensure that the power of test was 80% and able to detect differences in the means [27].

All lipidomic datasets (initially and after each variable reduction step for both plasma and erythrocytes) were preprocessed in the same fashion before employing statistical analysis (TIC normalized, Log-transformed and Pareto-scaled). Principal component analysis (PCA) score plots in both Metaboanalyst and SIMCA were used for data overview. The first filtration step was *t*-test for lipid significance (alpha = 0.01) on lipids between patients and controls performed in Metaboanalyst. Only lipids with FDR *p*-value < 0.01 were retained in the data matrix. This reduced dataset was imported to SIMCA where Orthogonal Partial Least Squares Discriminant Analysis (OPLS-DA) was performed with the goal to separate patients and controls. The rule to assess the lipid significance in this analysis was its OPLS-DA VIP score > 1. Hierarchical clustering analysis using Ward clustering algorithm and Euclidean distances options (HCA) was performed in Metaboanalyst. This further reduced final dataset was subjected to the above-mentioned univariate statistics. The differences between groups were evaluated using ANOVA with Fisher´s post hoc test. The statistical pipeline is shown in Appendix A.

## 3. Results

### 3.1. Oxidative Stress Markers

Oxidative burden and the defense capacity of plasma and erythrocyte samples was estimated according to several variables including direct oxidative stress biomarkers (MDA, protein carbonyls), activity of antioxidant enzymes (GSH-Px, Cu/ZN-SOD), non-enzymatic oxidants (GSH concentration, TAC) or redox status (GSH/GSSG ratio) (Table 2).

In both FO/– and FO/OO groups, we found an elevated MDA content, a marker of lipoperoxidation, in plasma and erythrocytes compared to controls. Increased protein carbonyl content in plasma (marker of protein oxidative stress) and low GSH/GSSG ratio in erythrocytes (redox status indicator) were detected only in FO/– group. In the FO/OO group, these parameters were normalized to the level observed in controls.

### 3.2. Lipidomic Analysis: Plasma

In plasma, we detected 592 MS/MS confirmed lipid signals. After filtering out all signals detected both in plasma samples and in lipid emulsions, 102 lipids remained. These compounds could be considered as products of intermediary metabolism and were further considered for the evaluation. First, we compared all patients versus controls. A principal component analysis (PCA) scores plot (Figure 1A) unraveled two well-separated clusters of patients and controls. Then, we performed a *t*-test that filtered out 81 significant lipids based on FDR *p*-value < 0.01 to be used in the next step of statistical analysis. Using supervised multivariate analysis (OPLSDA model with parameter of described variance R2 0.943 and predictive ability Q2 0.888) we identified 30 metabolites with VIP score ˃ 1.0 that significantly contribute to the separation of PN-dependent patients and controls. A hierarchical clustering analysis performed on this subset of lipids unraveled two main clusters, one of them consisting of controls plus three FO/OO subjects, the other formed by both FO/– and FO/OO subjects (Figure 1B). We did not find any clustering according to the OO presence or absence in patients provided FO.

Next, we performed ANOVA followed by Fisher´s post hoc test on this lipid subset (VIP score ˃ 1.0, *n* = 30) to unravel differences in lipid composition among groups (Appendix A). The abundance of all 30 lipids was significantly different in control vs. both FO/– and FO/OO groups, respectively. FO/– and FO/OO significantly differed in 15 lipid species. The distribution pattern of these lipids among the groups is shown in Figure 2. A common marker of both FO/– and FO/OO groups was high content of five fatty acyl esters of hydroxy fatty acids (FAHFAs). FO/– and FO/OO group significantly differed in the abundance of three lipid classes: (i) phospholipids containing mostly EPA and DHA (FO/– ˃ FO/OO); (ii) plasmanyls and (iii) natural antioxidants plasmalogens (FO/– ˂ FO/OO).

Taken together, our data show that plasma lipidome of PN-dependent patients differs from controls, but also between FO/– and FO/OO groups. Lipids contributing to the differences between groups the most are FAHFAs, plasmalogens, ω-3 PUFA containing phospholipids and lysophospholipids.

### 3.3. Lipidomic Analysis: Erythrocytes

In erythrocytes, we detected 195 MS/MS confirmed lipid signals. A PCA score plot created from this set of lipids documents separation of patient and control lipidomes with the exception of four samples from FO/OO, which were clustered with controls (Figure 3A). As in the plasma analysis, the *t*-test was performed and reduced the count of features to 138 lipids. Multivariate analysis (OPLS-DA with R2 0.966 and Q2 0.881) identified 59 lipid compounds (VIP score ˃ 1.0) that can distinguish patient and control cohorts. Hierarchical clustering analysis (Figure 3B) unraveled two main clusters: controls together with four subjects from the FO/OO group on one side and majority of patients on the other side. In the patient cluster, we observed no particular tendency to the separation of FO/– and FO/OO groups.

The subset of 59 lipids selected by OPLS-DA was subjected to ANOVA followed by Fisher´s post hoc test and 58 lipids were significantly different between at least two groups. The abundance of 55 and 52 lipid species was significantly different between controls and FO/– or FO/OO groups, respectively. Importantly, FO/– and FO/OO significantly differed in 33 lipid species (Appendix A). The distribution pattern of these lipids among the groups is shown in Figure 4. The lipids distinguishing controls from PN-dependent patients are phospholipids mostly containing EPA- and DHA-derived FA, acylcarnitines, ceramides, plasmalogens and plasmanyls. Acylcarnitines and ceramides content was similar in both FO/– and FO/OO groups and elevated compared with controls. Omega-3 PUFA-containing phospholipids were increased in both patient groups compared with controls but the abundance of most of them was significantly higher in FO/– compared with FO/OO. Plasmanyls and plasmalogens, other than those containing ω-3 PUFAs, were significantly decreased in patients compared with controls, however this effect was more pronounced in FO/– than in FO/OO.

In conclusion, our data demonstrate the effect of PN lipids on erythrocyte lipidome. The administration of PN mixtures containing OO translates into less pro-oxidant lipidome composition.

### 3.4. Correlation Analysis on MDA-Lipidome Level

We calculated the Spearman’s correlation coefficient to assess possible correlations between VIP lipids (score ˃1.0) and markers of oxidative stress (MDA concentration) in plasma and erythrocytes. In plasma, we found statistically significant correlations for 26 out of 30 VIP lipids, with adjusted p-values of mostly less than 0.001. MDA concentration was positively correlated with ω-3 PUFA-containing phospholipids, lysophospholipids and FAHFAs (18:3/18:1._7.32; 18:3/18:1._7.48; 14:1/22:2; 16:1/18:3) and negatively correlated with plasmanyls and plasmalogens (Figure 5A). In erythrocytes, the association between MDA concentration and lipidome composition was less pronounced. Only 14 out of 58 VIP lipids significantly correlated with MDA content, with adjusted *p*-values ranging from 0.03 to 0.049 (Figure 5B). We did not observe any correlation between ω-3 PUFA-containing lipids and MDA concentration in erythrocytes. These data indicate that the level of the oxidative stress and the lipidome composition are interconnected in both plasma and erythrocytes, but in erythrocytes, this association is less pronounced.

## 4. Discussion

### 4.1. Oxidative Stress in PN-Dependent Patients

Our study reveals a pro-oxidative state in patients receiving long-term ω-3 PUFA-rich PN as well as a partial (at least) oxidative stress-ameliorating effect of OO supplementation. These results seem to support the pro-oxidative role of ω-3 PUFAs in the PN context, which is in line with the study of Lavoie et al. in newborn guinea pigs showing pro-oxidative effect and induction of hypo-alveolarization following exaggerated apoptosis^18^. In contrast, Zang et al. observed alleviation of pro-oxidative state and improvement of liver markers in children with PN dependency^19^. However, comparing the results of individual studies is difficult as they vary in duration of PN administration, FO doses and inflammatory status of patients. The age of patients should also be taken into account regarding oxidative stress, because the adult PN-dependent patients are frequently elderly individuals (mean age is 60 and 66 years in FO/– and FO/OO groups, respectively). Ageing is associated with the accumulation of oxidatively modified lipids and proteins as well as a reduced capacity of specific antioxidant systems [28,29,30]. Therefore, we carefully selected the age-matched controls to compensate for the age-related bias.

We demonstrated the increased lipoperoxidation in all PN-dependent patients as evidenced by a higher MDA concentration in plasma and erythrocytes. In plasma, we observed a positive correlation between MDA concentration and ω-3 PUFAs containing lipid species, which further supports the pro-oxidative effect of ω-3 PUFAs. Moreover, FO, rich in ω-3 PUFA, and SO, rich in ω-6 PUFA, lipid emulsions for PN often contains OO. As reviewed recently by Cai et al. [8], olive oil and its primary constituent, ω-9 monounsaturated oleic acid, is associated with less lipid peroxidation compared with other lipid emulsions [31,32,33,34,35]. In our study, patients receiving FO/– emulsion exhibited increased protein oxidative stress (protein carbonyls) in plasma or compromised redox status (GSH/GSSG) in erythrocytes while these parameters were normal in those receiving FO/OO emulsion. Therefore, our data indicate that OO may partly counteract the pro-oxidative effect of FO in PN mixtures.

### 4.2. Functional Lipidomics and Biomarkers of Oxidative Stress

Lipids and their metabolites are essential components of biological systems. The detailed determination of their altered levels may yield sensitive biomarkers of ongoing pathological processes. Our findings regarding lipidome composition may contribute to the explanation of an OO protective effect.

Not surprisingly, we found the increased amount of ω-3 PUFAs-containing phospholipids and lysophospholipids in plasma and erythrocytes for all patients receiving ω-3 PUFAs in PN, but this elevation was significantly more pronounced in FO/– compared with FO/OO group.

We also observed the significantly decreased content of plasmalogens in PN patients. Plasmalogens are unique phospholipid species containing vinyl ether moiety at the *sn-1* position and highly unsaturated fatty acids at the *sn-2* position of the glycerol backbone [36]. The vinyl–ether bond is preferentially oxidized when exposed to free radicals [37]. Importantly, oxidative products of plasmalogens are unable to further propagate lipid peroxidation [38]. Therefore, plasmalogens are referred as “endogenous antioxidants”. In line, we observed a significant reduction of plasmalogen concentrations and their negative correlations with MDA concentration in patient plasma, which indicates the presence of oxidative stress and may suggest a lower capacity to cope with peroxidation products. Within our patient population, the plasmalogen depletion was more severe in the FO/– compared to the FO/OO cohort. We suppose that the lower plasmalogen content in FO/– group corresponds with its higher susceptibility to oxidative damage (protein carbonyls, GSH/GSSG ratio). In line, recent studies revealed the association between low plasmalogen levels in the blood and metabolic diseases associated with oxidative stress and chronic inflammation, i.e., type 2 diabetes, atherosclerosis, autoimmune diseases or neurodegenerative diseases [37,39,40].

Another characteristic feature, discriminating all PN patients from controls, is an elevated content of FAHFAs. Only recently, this novel class of endogenous lipokines, branched fatty acid esters of hydroxy fatty acids, was identified [41]. The biological function of FAHFAs is far from being completely understood. The first reports show their potential beneficial effects in glucose homeostasis [41] and suppressing inflammation [42]. In our study, FAHFAs derived from palmitoleic, oleic, arachidonic and tetradecanoic acids (fatty acyl moiety) and α-linolenic or dodecadienoic acids (hydroxyl fatty acid moiety) were elevated in plasma of all patients compared to controls. Our study design does not allow for the identification of the mechanism leading to the FAHFA elevation in PN-dependent patients and thus further research is needed to unravel its role in PN-associated metabolic adaptation.

### 4.3. Limitations of the Study

There are several limitations in the interpretation of the observed data, i.e., the heterogeneity of patients in causes of intestinal failure as well as significant inconsistencies in dose and duration of PN treatment. Furthermore, it is important to mention that PN-dependent patients exhibited higher CRP concentration, which could explain the increased oxidative stress markers independently from the type of lipid emulsion. Therefore, we cannot attribute our observations solely to the administration of ω-3 PUFAs in the nutrition mixture. Further research is needed to elucidate this issue.

## 5. Conclusions

We demonstrated the evidence of increased oxidative stress in patients receiving long-term PN with FO lipid emulsions compared to healthy controls. The oxidative stress was partly alleviated by OO supplementation. Moreover, the higher FAHFAs and lower plasmalogen levels were observed in plasma and erythrocyte lipidome of PN patients compared to controls. The decrease in plasmalogen concentration was more pronounced in the FO/– group, suggesting the protective effect of OO on oxidative stress.

## Figures and Tables

**Figure 1 nutrients-12-02351-f001:**
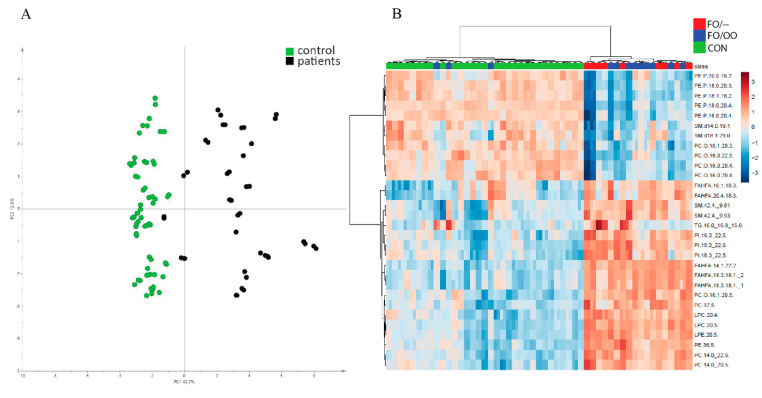
Plasma lipidome. (**A**): Principal Component Analysis (PCA) score plot based on all confirmed plasma lipids (*n* = 102) showing two separated clusters of control samples as green points and patient samples as black points. (**B**): Hierarchical clustering analysis of individual control and patient plasma samples according to all lipids with VIP score ˃1.0 (*n* = 30). The most distant clusters are a group consisting of controls and three FO/OO subjects on one side and all FO/– patients plus ten FO/OO patients on the other side. Within the later cluster, no special clustering according to the presence/absence of OO was seen. PCA principal component analysis; OO, olive oil; FO, fish oil.

**Figure 2 nutrients-12-02351-f002:**
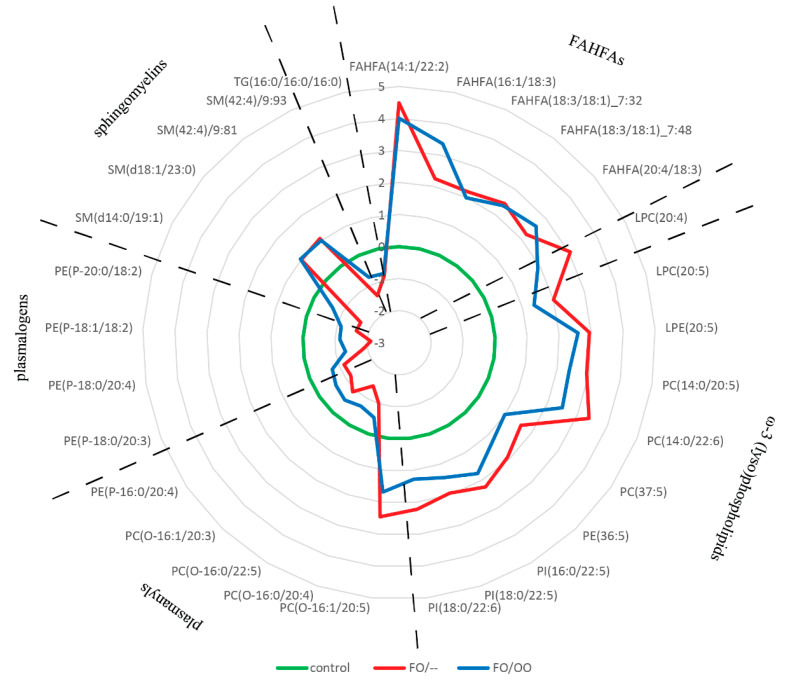
Radar chart showing distribution pattern of VIP lipids (score ˃1.0) in plasma. Data are expressed as log(2)FC over control. Only lipids surviving *t*-test and OPLS-DA selection (*n* = 30) were included into this analysis. Individual lipid species are grouped according to the lipid classes. For statistical significance calculated by ANOVA with Fisher´s post hoc test, see Appendix A. FC, fold change; OO, olive oil; FO, fish oil; PUFA, polyunsaturated fatty acids; FAHFAs, fatty acyl-esters of hydroxy-fatty acids.

**Figure 3 nutrients-12-02351-f003:**
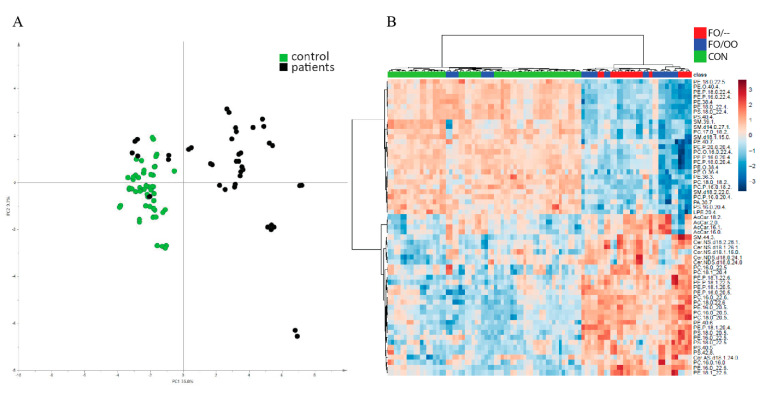
Erythrocyte lipidome. (**A**): Principal Component Analysis (PCA) score plot based on all confirmed erythrocyte lipids (*n* = 195) showing two separated clusters of control samples as green points and patient samples as black points. The exceptions are four samples from FO/OO group, which cluster with controls. (**B**): Hierarchical clustering analysis of individual patient erythrocyte samples according to all lipids with VIP score ˃1.0 (*n* = 59). There are two main clusters, i.e., patients on one side and controls together with four subjects from the FO/OO group on the other side. In the patient cluster, there is no particular tendency to the separation of FO/– and FO/OO groups. PCA, principal component analysis; OO, olive oil; FO, fish oil.

**Figure 4 nutrients-12-02351-f004:**
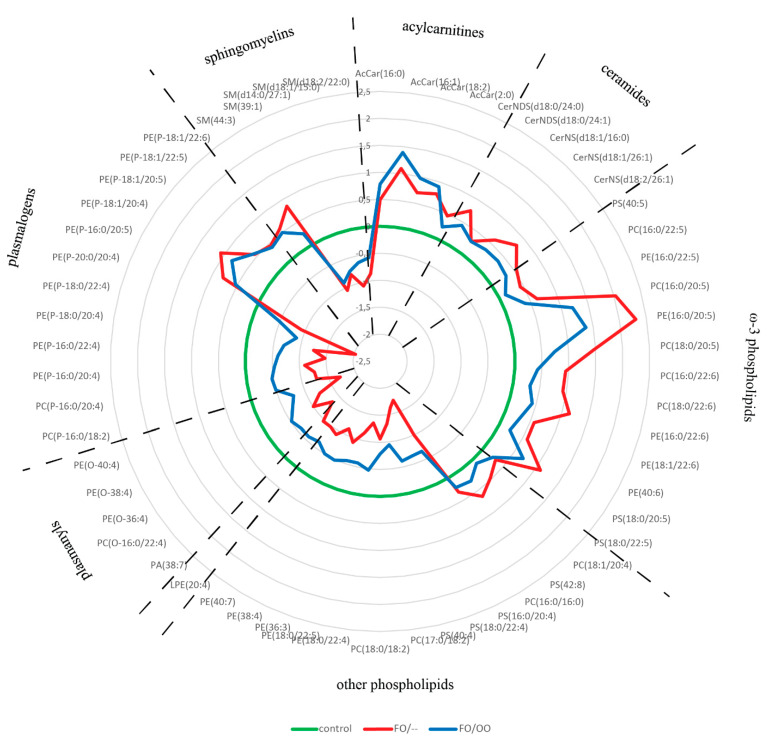
Radar chart showing distribution pattern of VIP lipids (score ˃1.0) in erythrocytes. Data are expressed as log(2)FC over controls. Only lipids surviving *t*-test, OPLS-DA selection and ANOVA test (*n* = 58) were included into this analysis. Individual lipid species are grouped according to the lipid classes. For statistical significance calculated by ANOVA with Fisher´s post hoc test, see Appendix A. FC fold change. OO, olive oil; FO, fish oil.

**Figure 5 nutrients-12-02351-f005:**
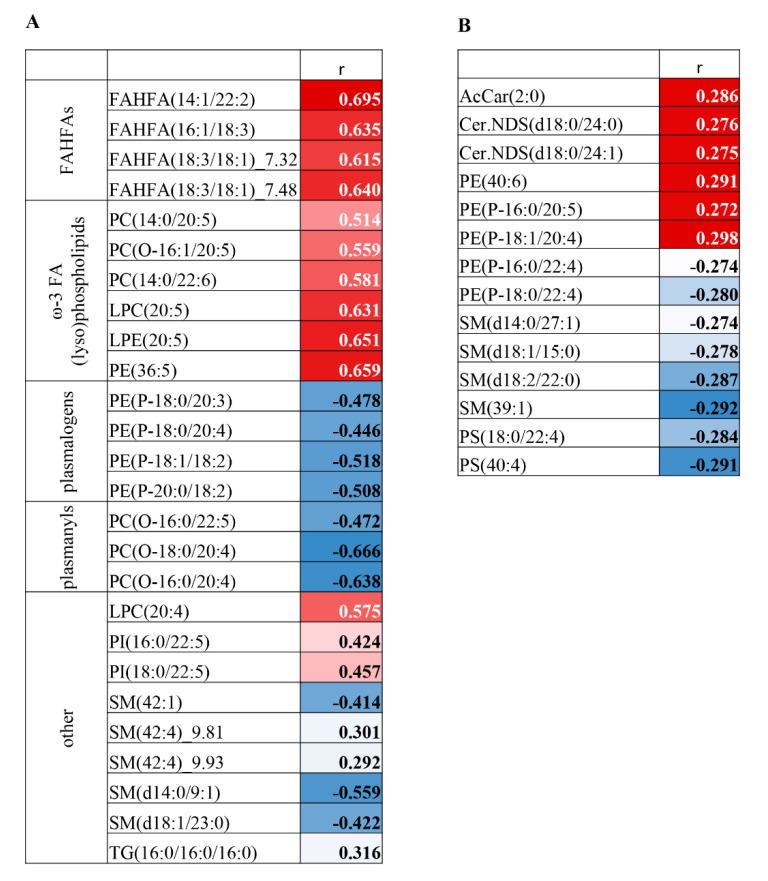
Correlation between MDA concentration and VIP lipids in plasma (**A**) and erythrocytes (**B**). The correlation was calculated as Spearman coefficient; only statistically significant values are shown. Blue color indicates negative and red color positive correlation. MDA, malondialdehyde; PUFA, polyunsaturated fatty acids; FAHFAs, fatty acyl-esters of hydroxy-fatty acids.

**Table 1 nutrients-12-02351-t001:** Cohort characteristics.

	Control	FO/–	FO/OO
sex (F/M)	20/10	7/2	10/3
age (years)	64 (40;80)	60 (32;73)	66 (42;81)
BMI	25.7 (19.3;30.2)	25 (18;28)	21 (16;30)
time on PN (month)	N/A	40 (2;113)	56 (7;124)
CRP (mg/l)	0.9 (0.1;6.5)	1.6 * (0.3;7.3)	1.1 * (0.9;13.6)
**diagnosis**			
SBS I	N/A	7	3
SBS II + III	N/A	2	4
other	N/A	0	6
**parenteral nutrition**			
i.v. energy (kcal/day)	N/A	940 (568;1575)	1144 (545;1705)
lipids (g/day)	N/A	20 (5.7;50)	32.2 (5.7;56.3)
soybean oil (g/day)	N/A	8.0 (2.3;20)	9.7 (1.7;16.9)
olive oil (g/day)	N/A	0.0	8.1 ^‡^ (1.4;15.2)
MCFA (g/day)	N/A	10.0 (2.9;25)	9.7 (1.7;16.9)
fish oil (g/day)	N/A	2.0 (0.6;5.0)	4.6 (0.9;8.4)
amino acids (g/day)	N/A	50.0 (28.6;71.5)	53.6 (28.6;108.6)
glucose (g/day)	N/A	138 (71;250)	134 (36;200)
**lipid metabolism**			
TAG (mmol/L)	1.5 (0.8;3.6)	1.6 (1.1;2.6)	1.2 (0.5;2.4)
total cholesterol (mmol/L)	5.3 (3.6;7.6)	3.3 (2.6;4.3)	3.8 (2.0;4.5)
HDL cholesterol (mmol/L)	2.2 (0.9;6.1)	1.1 (0.5;1.4)	1.2 (1.0;2.1)
LDL cholesterol (mmol/L)	2.0 (1.0;4.3)	1.6 * (0.8;2.3)	1.6 * (1.0;2.3)
**liver function tests**			
bilirubin (µmol/L)	9.9 (4.8;27.3)	13.0 (6.7;33.1)	9.5 (5.1;88.1)
AST (µkat/L)	0.4 (0.3;1.8)	0.5 (0.4;1.4)	0.4 (0.3;2.5)
ALT (µkat/L)	0.6 (0.4;1.1)	0.8 (0.4;4.1)	0.4 (0.3;1.0)
ALP (µkat/L)	1.3 (0.7;2.1)	2.2 (1.4;3.1)	1.9 (1.1;2.7)
GGT (µkat/L)	0.4 (0.2;1.8)	0.7 (0.2;2.9)	0.6 (0.2;2.2)
albumin (g/L)	42 (35;47)	43 (39;48)	44 (36;53)
**other**			
blood count	4.4 (3.9;5.2)	3.9 (3.4;4.8)	4.3 (2.8;5.1)
uric acid	253 (180;287)	268 (179;573)	236 (116;353)
total protein content	70 (58;81)	73 (68;83)	71 (67;81)

ALT, alanine aminotransferase; ALP, alkaline phosphatase; AST, aspartate aminotransferase; BMI, body mass index; CRP, C-reactive protein; F, female; GGT, gamma-glutamyl transferase; HDL, high density lipoprotein; LDL, low density lipoprotein; M, male; MCFA, medium chain fatty acids; PN, parenteral nutrition; TAG, triacylglycerol. Data are given as a median (min; max). * *p* ˂ 0.05 vs. control; ^‡^
*p* ˂ 0.05 vs. FO/–.

**Table 2 nutrients-12-02351-t002:** Oxidative stress markers in erythrocytes and plasma.

		Control	FO/–	FO/OO
	**oxidative stress markers**			
**erythrocytes**	MDA (nmol·g Hb^−1^)	7.4 (5.1;10.9)	14.3 *** (7.0;31.9)	13.8 *** (5.0;35.9)
	protein carbonyls (nmol·mg Hb^−1^)	0.4 (0.0;1.1)	0.7 (0.3;1.2)	0.4 (0.3;1.1)
	**non-enzymatic oxidants**			
	antioxidant capacity	16.0 (10.0;21.6)	15.6 (9.2;18.4)	15.6 (11.6;21.6)
	GSH	1430 (847;2398)	1187 (789;2059)	1736 (947;2215)
	GSSG	57.5 (29.1;97.8)	64.3 (33.3;116.5)	74.1 (38.1;109.6)
	**redox status**			
	GSH/GSSG	25.3 ^†^ (11.0;44.1)	17.7 * (15.1;22.3)	24.8 ^†^ (15.7;41.4)
	**oxidative stress markers**			
**plasma**	MDA (µM)	1.4 (1.0;1.9)	2.3 *** (1.6;3.4)	2.1 *** (1.2;3.4)
	protein carbonyls (nmol mg Hb^−1^)	1.8 ^†^ (0.0;4.2)	2.9 * (1.0;6.3)	1.3 ^†^ (0.5;5.3)
	**non-enzymatic oxidants**			
	antioxidant capacity	59.6 (41.6;79)	55.2 (41.2;114)	47.2 (37.6;67.2)
	**antioxidant enzymes**			
	SOD (U/ml)	1.8 (1.1;2.2)	1.5 (1.1;2.0)	2.0 (1.3;2.3)
	GSH-Px (µM NADPH/min/ml)	330 (280;360)	375 (330;410)	370 (310;440)

MDA, malondialdehyde; Hb, hemoglobin; GSH, glutathione; GSSG, glutathione disulfide; SOD, superoxide dismutase; GSH-Px, glutathione peroxidase; NADPH, nicotinamide adenine dinucleotide phosphate; SO, soybean oil; OO, olive oil; FO, fish oil. Data are given as a median (min;max). * *p* ˂ 0.05, *** *p* ˂ 0.001 vs. control; ^†^
*p* ˂ 0.05 vs. FO/–.

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
