# Peer review of "The ω-3 Polyunsaturated Fatty Acids and Oxidative Stress in Long-Term Parenteral Nutrition Dependent Adult Patients: Functional Lipidomics Approach"

_nutrients, 2020, doi:10.3390/nu12082351_

Round 1

Reviewer 1 Report

A very comprehensive and solid study about ω-3 PUFAs and oxidative stress in adult patients dependent on parenteral nutrition. The authors used an unbiased lipidomics approach to show the protective effect of ω-3 PUFAs and olive oil in vivo. My only concern is that the cohort of patients is relatively small, but we all know it is very hard to establish a large cohort in this situation.

Minor suggestion 1: if the authors can cite one review article from Dr.Klek (PMID: 26959070), it could give readers a whole picture of ω-3 PUFAs and parenteral nutrition.
Minor suggestion 2: The resolution of fig1 and fig 3 is too low, please upload more clear heat-maps.

Author Response

Response to reviewer 1

We thank the reviewer for time and effort devoted to our manuscript and for positive evaluation.

Minor suggestion 1: if the authors can cite one review article from Dr.Klek (PMID: 26959070), it could give readers a whole picture of ω-3 PUFAs and parenteral nutrition.

We added the proposed reference to the Introduction (pg. 3, line 61).

Minor suggestion 2: The resolution of fig1 and fig 3 is too low, please upload more clear heat-maps.

Figures were uploaded as separate files in better resolution (300 dpi).

Reviewer 2 Report

The overall gal of this paper is rather interesting but, this paper lacks clarity and some parts are difficult to understand.  I would suggest a careful reading: e.g. lines 41, 51-52, 158, etc.

The introduction is correct (not great, the information found in this introduction are quite basic).  The methods are clearly described.  I am not sure that table should be present in the results section or more likely in the method section.  The results are correctly presented. 

The discussion is poor.  For instance the first sentence is difficult to understand and does not make sense to me : “Our study demonstrates a rather pro-oxidative state in patients receiving ω-3 PUFA-rich PN as 261 well as at least partially protective effect of olive oil supplementation”

                1 – demonstrates : quite strong

                2 - A rather pro-.. : quite weak

                3 – a partly protective effect : on what ?

The next sentence is not easier to understand.  There is in it references to other papers but no details are given.  Please put some efforts on this part of the manuscript.

Author Response

Response to reviewer 2

We thank the reviewer for considering our manuscript and for valuable comments. We hope that we addressed all of the issues raised and we believe that it led to the significant improvement of the manuscript.

  1. The text, especially Abstract, Discussion, Limitations of the study and Conclusions, were revised and reformulated. The changes are underlined in red.
  2.  
  3. The description of cohort characteristics and Table 1 was moved from the section “Results” to the section “Methods”.